# Non-Invasive Nasal Discharge Fluid and Other Body Fluid Biomarkers in Alzheimer’s Disease

**DOI:** 10.3390/pharmaceutics14081532

**Published:** 2022-07-22

**Authors:** Da Hae Jung, Gowoon Son, Oh-Hoon Kwon, Keun-A Chang, Cheil Moon

**Affiliations:** 1Department of Brain Sciences, Graduate School, Daegu Gyeongbuk Institute of Science and Technology (DGIST), Daegu 42988, Korea; dhjung@dgist.ac.kr (D.H.J.); gowoonlizson@gmail.com (G.S.); 2Convergence Research Advanced Centre for Olfaction, Daegu Gyeongbuk Institute of Science and Technology (DGIST), Daegu 42988, Korea; jxstigma0114@dgist.ac.kr; 3Neuroscience Research Institute, Gachon University, Incheon 21565, Korea; keuna705@gachon.ac.kr; 4Department of Pharmacology, College of Medicine, Gachon University, Incheon 21936, Korea; 5Neuroscience of Health Sciences and Technology, Gachon Advanced Institute for Health Sciences and Technology, Gachon University, Incheon 21936, Korea

**Keywords:** Alzheimer’s disease, nasal discharge fluid, body fluid, biomarker, diagnosis, non-invasive, peripheral

## Abstract

The key to current Alzheimer’s disease (AD) therapy is the early diagnosis for prompt intervention, since available treatments only slow the disease progression. Therefore, this lack of promising therapies has called for diagnostic screening tests to identify those likely to develop full-blown AD. Recent AD diagnosis guidelines incorporated core biomarker analyses into criteria, including amyloid-β (Aβ), total-tau (T-tau), and phosphorylated tau (P-tau). Though effective, the accessibility of screening tests involving conventional cerebrospinal fluid (CSF)- and blood-based analyses is often hindered by the invasiveness and high cost. In an attempt to overcome these shortcomings, biomarker profiling research using non-invasive body fluid has shown the potential to capture the pathological changes in the patients’ bodies. These novel non-invasive body fluid biomarkers for AD have emerged as diagnostic and pathological targets. Here, we review the potential peripheral biomarkers, including non-invasive peripheral body fluids of nasal discharge, tear, saliva, and urine for AD.

## 1. Introduction

Alzheimer’s disease (AD) is an irreversibly progressive neurodegenerative disease afflicting the elderly, accompanied by devastating cognitive and memory impairment caused by characteristic neuronal and synaptic loss and cortical and hippocampal atrophy. It is hallmarked by the accumulation of extracellular amyloid plaques and intracellular neurofibrillary tangles. The underlying mechanisms contributing to the development of the disease remain elusive and controversial. Despite the advancement in understanding the mechanism of pathogenesis, clinical trials have been unsuccessful and provided no relief from disease progression, only slowing the progression [1]. Recent FDA-approved anti-amyloid therapy aducanumab highlights that it is effective for patients with very mild, biomarker-proven AD [2,3]. Therefore, there is an urgent need to develop a more accessible biomarker screening test using less invasive and cost-effective body fluid biomarkers. These diagnostics will serve as the first line of effective AD therapies before extensive pathophysiological brain devastation occurs.

Currently, AD diagnosis involves a combination of neuroimaging techniques, detailed clinical review of family history, neuropsychological test results, and laboratory assay results [4,5,6,7]. In the field of early AD diagnosis, the biomarkers using cerebrospinal fluid (CSF), blood or neuroimaging, using magnetic resonance imaging (MRI) or positron emission tomography (PET), are being rapidly developed. However, the high cost of test procedures, potential complications, need for a specialist, requirement for high-performance equipment, lack of standardized cross-institution protocols, and inconsistencies in test result interpretation hinder the accessibility of these screening tests to individuals in economically disadvantaged areas or remote geographical regions [8]. These factors hinder easy accessibility to routinely approaching these diagnostic tests, and therefore, patients at risk may miss the opportunity for a timely and accurate diagnosis and pharmacological intervention.

In an attempt to overcome these shortcomings, a rising number of biomarker profiling research using non-invasive peripheral body fluids such as nasal discharge, saliva, urine, and tear has shown the potential to capture the pathological changes in the patients’ bodies. Of 15,445 AD biomarker-related articles published in the last 10 years, 355 articles related to peripheral body fluid biomarker were retrieved when searched in PubMed, and 71% of them were published in the last five years, showing that there is a growing awareness about the importance of peripheral body fluid biomarkers in AD research. In view of this, this review aims to provide an overview of recent contributions in the field of non-invasive body fluid biomarkers and to shed new light on the potential of non-invasive bodily fluid biomarkers. Few previous studies have provided standardized guidelines for interpreting diagnosis criteria or biomarker source origins regarding AD biomarker research papers. Taken together, we will address this issue and formulate guidelines for reading AD biomarker research papers.

## 2. Clinical Diagnosis of AD

The standard diagnostic criteria for AD were first established in 1984. First, the diagnosis of Alzheimer’s disease was initiated by the National Institute of Neurological Disorders and Stroke (NINDS) and the Alzheimer’s Disease and Related Disorders Association (ADRDA) [9]. The criteria were compatible with the Diagnostic and Statistical Manual of Mental Disorders (DSM-III) [10]. The NINCDS-ADRDA and DSM-III evaluated the cognitive impairment by AD and dementia syndrome, which are summarized in Table 1. Clinical diagnostic criteria of possible and probable AD were made based on neuropsychological tests. The criteria for diagnosing definite AD involved clinicopathologic investigations, meaning histopathological evidence from microscopic examination obtained from autopsy or biopsy.

Next, the National Institute on Aging-Alzheimer’s Association (NIA-AA) workgroups updated the diagnostic guideline for AD [11,12]. Revised diagnostic criteria provide earlier identification of AD progression, including preclinical and mild cognitive impairment (MCI) stages. Including the NIA-AA guideline, revised DSM-V is also incorporated into the classification of disease progression [13]. The criteria involve neuropsychological tests such as Mini-Mental State Examination (MMSE), Clinical Dementia Rate (CDR), and Global Deterioration Scale (GDS), and the postmortem examination confirms AD pathology, amyloid plaques, and neurofibrillary tangles. These provide an initiative research framework underlying pathologic processes that can be reported by postmortem examination. However, AD diagnosis’s significance shifted to observing the disease progression in living people rather than defining the consequences because the current treatment is ineffective after severe neurodegeneration.

Recent AD research has led to the biomarker analysis in vivo for timely and appropriate intervention before a severe brain injury. As a result, the research framework was updated to examine the AD pathologic processes [14]. In 2016, Jack et al. proposed the A/T/N classification as a means of evidencing the biological state of AD [15]. This classification is divided into binary categories, either positive or negative, and this system provides an improved understanding of the sequence of events in the AD continuum [14]. This framework includes A, the status of amyloid-β (Aβ) PET or CSF Aβ_42_; T, tau PET or CSF phosphorylated-tau (P-tau); N, neurodegeneration or neuronal injury measured by neuroimaging or CSF total-tau (T-tau). The presence of each A/T/N biomarker profiles the pathologic change with or without AD. Table 2 summarizes clinical AD stages based on the NIA-AA guideline and the results of neuropsychological batteries.

## 3. Conventional AD Body Fluid Biomarkers

A biomarker indicates the change of cells and tissues, which illustrates altered body conditions [16]. In the clinics, biomarkers represent the pathological status and monitor the disease progression. Appropriate methods to detect disease biomarkers in living patients are important for establishing early intervention times and evaluating clinical therapeutic efficacy. The characteristics of an ideal biomarker are outlined in Table 3.

Given that AD is progressive and incurable, an ideal method to detect AD biomarkers is required for AD-specific early detection, economic accessibility, and non-invasive sample collection [17]. The neuropsychological test is a worldwide and classic method to identify cognitive impairment, but multiple profiling is demanded to confirm AD from other neurodegenerative diseases. Although brain imaging (e.g., fMRI, FDG-, amyloid-, and tau-PET) observes disease-specific pathophysiology, the patients may find it difficult to perform neuroimaging tests due to their repeatedly high costs. Recently, observing amyloid-β and tau as biomarkers in CSF and plasma are approved by NIA-AA [14]. In addition to CSF and blood, body fluids that can closely reflect the patient’s pathological condition have been studied. Figure 1 illustrates body fluids that have the potential to be matrices of biomarker detection.

### 3.1. CSF

CSF is a clear and colorless body fluid in the subarachnoid space and circulates within the ventricular system of the brain and spinal cord to supply nutrients and chemicals, remove waste products, and provide the brain immunological protection and mechanical support [18,19]. CSF is produced in the choroid plexus of the brain’s ventricles and reabsorbed into venous sinus blood via the arachnoid granulations. The total volume of CSF is approximately 125–160 mL. CSF is replaced four to five times and regenerates about 500 mL every day [20]. Since CSF reflects biochemical and environmental changes within the central nervous system, CSF is an ideal and useful candidate for detecting potential neuropathology biomarkers [21,22]. CSF is usually obtained by a procedure called a lumbar puncture. The standardized collection protocol of lumbar puncture is carried out under sterile conditions by inserting a needle into the subarachnoid space between the third and fifth lumbar vertebrae [23]. The collection protocol is summarized in Table 4.

NIA-AA and International Working Group (IWG) 2 have recognized the significance of CSF biomarkers, including Aβ_42_, T-tau, and P-tau, and incorporated them into diagnostic criteria for AD and MCI [24,25,26]. In AD, Aβ_42_ concentration and Aβ_42_/Aβ_40_ ratio are reduced, and T-tau and P-tau concentrations increase in CSF [18,27,28,29]. CSF-related core AD biomarker changes are summarized in Table 5.

Although lumbar puncture is the most common and recommended method for CSF collection, there are some issues: lumbar puncture results in discomfort and pain due to the larger and longer needle and the possibility of CSF contamination by anesthesia. In addition, it is very difficult and expensive to perform the procedure on the subjects repeatedly [23].

### 3.2. Blood (Plasma and Serum)

Blood plasma is the liquid component in which blood cells are suspended [30]. It delivers nutrients and oxygen to the cells and transports cellular metabolic products. It amounts to about 55% of the total blood volume and is mostly water. Blood plasma is circulated through the body via blood vessels by the pumping of the heart. Functions of blood plasma are maintenance of the blood pressure, pH, immunity, and transportation of electrolytes, nutrients, clotting factors, carbon dioxide, oxygen, other waste products, and excretory proteins.

Blood serum is blood plasma without clotting factors such as fibrinogens [30]. Blood serum includes all electrolytes, antibodies, antigens, hormones, and other substances, except white blood cells, red blood cells, platelets, and clotting factors [31,32]. Blood serum is obtained by coagulation, which allows for clotting of the blood. Both plasma and serum are commonly used for proteomic analysis.

Blood samples are collected by venipuncture, and the protocol commonly used is based on the Human Plasma Proteome Project (HPPP) by the Human Proteome Organization (HUPO) and the National Institute of Health [33,34]. Table 4 summarizes acquisition procedures for blood plasma and serum.

The conventional biomarkers of AD, such as Aβ_42_, Aβ_40_, T-tau, and P-tau, are commonly utilized as potential candidate screening molecules in blood samples because they can pass the blood–brain barrier (BBB) [35,36,37]. However, the BBB breaks down in AD patients, which leads to an accumulation of blood-derived neurotoxic proteins in the brain [38]. Blood-related core AD biomarker changes are summarized in Table 5.

Biomarkers for neuropathology from blood samples have been controversial because blood communicates with the brain through the BBB, lymphatic vessel, and lymphatic system, which indirectly interchange the materials and substances from the brain into blood, resulting in lower sensitivity and specificity than CSF [35,39,40,41].

### 3.3. Limitations of Current CSF and Blood AD Biomarkers

Although the core CSF and blood AD biomarkers reflect central pathological changes of the disease, current analyses have drawbacks: invasive procedure, high cost of test procedures, potential complications, between-institution differences in cut-off values, and inconsistencies in test result interpretation [42,43,44]. Furthermore, a plethora of studies have characterized the multifaceted nature of AD, highlighting the complexity of understanding the biochemical changes in the disease progression [45]. Due to these limitations, the accessibility of diagnosis assays is hindered, making the diagnosis belated, which adds cost to health care systems. Therefore, the development of novel biomarker detection in non-invasive body fluid is essential.

## 4. Novel Peripheral Body Fluid Biomarkers

Recent research has shown that other peripheral body fluids, such as nasal discharge, tear, saliva, and urine, may represent a potential source of biomarkers for neurodegenerative diseases. These peripheral body fluids have advantages over CSF or blood since the collection methods are less invasive and enable low-cost biomonitoring. Table 6 summarizes acquisition methods for peripheral body fluids.

### 4.1. Nasal Discharge

The occurrence of olfactory deficits, named anosmia or hyposmia, in AD has been characterized for decades, and often these deficits precede the cognitive decline [82,83,84,85]. Olfactory neuropathology is the cause of olfactory dysfunction, and structural and functional evidence supports this view, including abnormal APP processing and neuroinflammation [86,87,88,89,90]. The central olfactory processing regions, such as entorhinal and transentorhinal areas, olfactory bulb, and other medial temporal lobes, anatomically overlap with the regions involved in early AD pathology [82,91,92]. AD postmortem and antemortem studies revealed that the olfactory system shows classic AD hallmarks such as intracellular neurofibrillary tau tangles and amyloid plaques [93,94,95,96,97]. In particular, nasal discharge surrounds the olfactory system and captures the neuropathology occurring in the system, emerging as a potential matrix of fluid biomarkers.

Nasal discharge is a slippery and gelatinous fluid produced by mucous membranes in the olfactory mucosa. Nasal discharge is 95% water, glycoproteins, proteoglycans, lipids, proteins, and DNA. The purpose of nasal discharge is to protect the olfactory epithelium (OE) and the respiratory system by blocking the infections of pathogenic antigens. Nasal discharge fluids serve to humidify and clean inhale air and provide proteins of the innate immune system. Additionally, nasal discharge fluids trap and dissolve odorants for the olfactory receptor neurons.

Since the olfactory system is exposed to the external environment, the collection of nasal discharge fluid is easily accessible and non-invasive. In Table 6, we described several protocols for collecting samples [66,69,71]. Saline buffers that have a similar composition to human body fluids are used for nasal irrigation. Various techniques and devices have been developed to deliver saline to the nasal cavity, such as douche, spray, and nebulizer. Nasal irrigation can effectively relieve sinusitis caused by respiratory tract infections [98] and symptoms involved in allergic responses [99]. Additionally, to analyze proteomic studies of nasal discharge fluids for biomarkers, nasal discharge fluid obtained through nasal irrigation can provide valuable information. The second method of nasal discharge fluid collection is to use sinus packs. The method is non-invasive and reproducible. Several techniques, such as nasal lavage, brush, and scraping, have been known as collection methods that may influence the results [100]. Watelet et al. proposed a new technique to obtain nasal discharge fluids using sinus packs [69]. The authors confirmed the fluid quantity and protein concentration from the sinus packs and evaluated the feasibility and reproductivity of this technique. Thirdly, a nasal swab (or nasopharyngeal swab) is a method for collecting a sample of nasal discharge fluid from the back of the nose or throat. This method is commonly used to analyze the presence of markers of disease, organisms, and viral infection. Recently, a nasal swab has been used to diagnose COVID-19 [101]. 

Early studies identified the presence of amyloid-β peptide and amyloid precursor proteins (APP) in postmortem AD patients’ olfactory mucosa samples [94,102]. Aggregation of amyloid-β expression was detected in 71% of AD cases, 22% in normal cases, and 14% in other neurodegenerative disease cases [103]. Biopsy examination identified Aβ expression from the normal, MCI, and AD subjects [104]. Sampling human olfactory environment for AD-related research advanced from taking autopsy or biopsy samples to collecting human nasal discharge. A study collected nasal smears by swabbing from multiple nasal areas, such as the common nasal meatus, inferior concha, middle nasal meatus, and olfactory cleft [71]. Subsequent studies analyzed Aβ expression in nasal discharge fluid by immunoassay and proved that the level of oligomeric Aβ in nasal discharge was higher in AD than normal [67,105]. A 3-year longitudinal study by Yoo et al. confirmed that the presence of oligomeric Aβ could predict the cognitive decline progression [67].

Similarly, early studies used the histopathological method to detect T-tau in postmortem and antemortem samples of AD patients’ olfactory systems [95,106]. Later, immuno-histochemical studies demonstrated that tau pathology in the olfactory system correlated with AD pathology progression [93,107]. ELISA-analysis of nasal smear swabs indicated that P-tau/T-tau ratios were more significant in AD than control [71,108].

Non-core AD hallmark biomarkers have also been identified in the olfactory system. Expression of α-synuclein was identified in postmortem OE of AD sample [103]. A study showed increased microRNA-206 in AD patients’ OE through qRT-PCR [109]. Proteome analysis was done on nasal discharge samples from young, healthy groups and elderly groups, and identified a list of associated proteins with age variability [110]. However, little is known about the molecular machinery responsible for mucus proteome and its changes in neurodegenerative diseases. Table 7 summarizes the results of core AD biomarker studies from the olfactory system.

### 4.2. Tears

Tears have a high protein content and have been widely investigated for biomarker studies for ocular diseases and diabetes [113,114]. Major tear proteins, lipocalin-1 and lactotransferrin, are involved in the inflammatory and immune processes [113,115]. Researchers studying neurodegenerative diseases have also hypothesized that neuroinflammation could be reflected in tear proteins due to the extension of the central nervous system. Recent AD studies conducted tear analyses and discovered the potential of tear biomarkers in AD. 

The techniques of collecting tears were established in 1981 [72] and 1984 [73], and the protocols described in Table 6 are the most commonly used in tear proteomic investigations [74]. For proteomic analysis, tears can be collected using Schirmer’s tear strips and capillaries. The Schirmer strip is placed in the lower eyelid and allowed to absorb the tear for several minutes. The capillary tear is collected using sterile capillary tubes under the same conditions. However, to obtain aqueous humor samples, surgical treatment is required to use a fine needle [116], and vitreous humor samples are obtained by surgery vitrectomy [117].

Gijs et al. measured Aβ_42_ in tears using multiplex immunoassays and found the Aβ_42_ levels changed with increasing AD stage with an area under the curve (AUC) of 0.725 [118]. Other Aβ peptides, Aβ_38_ and Aβ_40_, were also detected in their subsequent study [119]. Recently, Wang et al. developed a biosensor and detected variable Aβ_42_ levels in different age groups of healthy participants [120]. Gijs et al. also analyzed T-tau, and its levels were also able to discriminate between AD patients and healthy controls with an AUC of 0.81 [118,119]. Quantitative proteomic results profiled that lipocalin-1, dermcidin, lysozyme-C, and lacritin can serve AD biomarkers [121]. One LC/MS evaluation identified elongation initiation factor 4E (elF4E) uniquely in AD tear samples, and a PCR-based analysis showed elevated total microRNA abundance in AD patients’ tears and especially higher microRNA-200b-5p levels in tears of AD patients compared to healthy controls [122]. Table 8 summarizes the results of core AD biomarker studies using tears.

### 4.3. Saliva

Saliva is an easily accessible, non-invasive body fluid containing 98% water containing electrolytes, proteins, peptides, hormones, sugar, epithelial cells, white blood cells, enzymes, and lysozymes [123]. The functions of saliva are the protection and maintenance of oral mucosa, digestion, the perception of taste, and the control of microorganisms [124,125]. Saliva is secreted from three major salivary glands, named the sublingual, submandibular, and parotid, and they are innervated by the cranial and facial nerves [126]. The direct innervation of the glossopharyngeal nerve through the otic ganglion suggests that saliva can be a promising candidate of biomarker source for assessing pathological physiologies of the nervous system [127].

Several various methods for collecting saliva have been described in the past years. In 2007, the World Health Organization and International Agency for Research on Cancer described the protocol for saliva proteomics [77]. The protocols for saliva collection depend on the specific categories of the saliva of interest, and these different methods are summarized in Table 6.

In the last few years, various studies have detected increased Aβ_42_ in AD patients’ saliva using sandwich and nanobead ELISAs. Bermejo-Pareja et al. analyzed saliva samples by immunoassays and identified a statistically significant increase in saliva Aβ_42_ levels in mild AD patients than normal control [128]. Subsequent studies similarly showed elevated Aβ_42_ levels in AD saliva samples [129,130,131]. In contrast to these findings, other results showed no detection of Aβ_42_ or Aβ_40_ in saliva with immunoassays [127,128,132]. On the other hand, a recent study demonstrated decreased Aβ_42_ level in AD patients’ saliva [133]. Some preliminary tau investigation was carried out in the 2010s, and Shi et al. reported an increased P-tau/T-tau ratio in patients with AD compared to healthy controls [127]. A subsequent study also confirmed this increased P-tau/T-tau ratio in AD versus healthy controls [134]. In contradiction with these findings, results demonstrated no significant difference in salivary T-tau between AD and mild cognitive impairment or healthy controls [135].

A growing number of studies have examined possible biomarker candidates other than Aβ and tau peptides. Lactoferrin, for instance, is a pleiotropic protein with several immunological properties, including antibacterial, antiviral, antioxidant, and anti-inflammatory functions [136,137]. The first investigations on salivary lactoferrin found a decreased lactoferrin level in AD patients’ saliva compared to healthy controls [138]. A more recent study supported this finding by comparing salivary lactoferrin levels with amyloid-PET neuroimaging data [139]. Acetylcholinesterase degrades acetylcholine neurotransmitters released into the synaptic cleft and terminates acetylcholine neurotransmission, and PET study results demonstrated decreased acetylcholinesterase catalytic activity in AD patients’ brain regions [140,141,142]. An initial report on the salivary acetylcholinesterase activity was carried out by Sayer et al. and showed a significant decrease in AD patients [143]. Further studies also suggested a reduced salivary acetylcholinesterase activity in AD patients [144,145]. Protein carbonyl levels result from protein oxidation, and multiple studies examined elevated protein carbonyls in brain regions of AD patients [146,147]. One study evaluated and identified protein carbonyl levels in saliva of AD patients and healthy controls [148]. Metabolomics is an emerging research technique used in various research fields to identify metabolites within a target sample. Yilmaz et al. analyzed saliva samples from healthy control, mild cognitive impairment sufferers, and AD patients. They profiled multiple metabolites that changed significantly in the saliva of MCI and AD patients compared to healthy controls [149]. Table 9 summarizes the results of core AD biomarker studies using saliva.

### 4.4. Urine

Urine contains thousands of proteins, mostly metabolic wastes, and is currently actively utilized to study pregnancy, aging, and kidney diseases [150,151]. Nevertheless, for many years urine has been neglected as a promising source of biomarkers for studying AD since there is little agreement on urine reflecting the changes occurring in the brain due to the BBB and glomerular filtration. However, several studies have indicated the potential of urinary biomarkers in neurodegenerative diseases, such as Parkinson’s disease and Alzheimer’s disease [152,153,154].

More importantly, urine collection does not require special equipment and can be repeated without discomfort to subjects. The Human Kidney and Urine Proteome Project (HKUPP) in 2005 and European Kidney and Urine Proteomics (EuroKUP) in 2008 were initiated to promote proteomics research, and they together have achieved the establishment of a standard protocol for urine collection and storage [78,79,80,81]. The collection method is summarized in Table 6.

Initial report on detecting Aβ peptide in the urine of AD patients was carried out by Takata et al. by Western blot analysis and suggested that monomeric Aβ level may reflect the severity of AD [155]. A key question raised by Takata et al. was that they could not pinpoint the origin of Aβ in urine. A recent study developed an indirect competitive ELISA to measure Aβ_42_ in human urine samples [156].

Several studies have profiled potential urine protein biomarkers for AD. One study identified 15 proteins using LC/MS-MS and validated three proteins, SPP1, GSN, and IFGBP7, by ELISA [157]. Higher urinary AD7c-NTP, Alzheimer-associated neural thread protein, was demonstrated in another study [158]. Watanabe et al. analyzed AD patients’ urine samples and profiled 109 proteomes differentially expressed in AD and healthy controls [159]. Their subsequent study showed that apolipoprotein C3 levels in AD patients’ urine samples were higher in the AD and MCI groups than healthy controls using ELISA [160]. Urine also contains many metabolites, reflecting the gut microbiome theory in neurodegeneration studies [161,162,163]. One study recently examined urinary metabolome using NMR spectroscopy and UHPLC-MS and built a model that could discriminate between AD and healthy age-matched controls [164]. Another study developed a new screening approach using LC/MS and proposed that lipid peroxidation compounds may be potential predictors of early AD [165]. In addition, many studies reported microRNA in human urine samples and demonstrated that urinary microRNAs are relatively stable under various storage conditions, supporting their utility as urinary biomarkers [166,167,168]. Table 10 summarizes the results of core AD biomarker studies using urine. 

### 4.5. Limitations and Future Perspectives of Novel Peripheral Body Fluid Biomarkers

Currently, a multitude of studies are discovering potential body fluid biomarkers to assist in assessing disease progression, developing treatments, and monitoring treatment efficacy. Nevertheless, there are substantial challenges in the validation and application of peripheral body fluid biomarkers in AD to clinical practice, and there is no single ideal peripheral body fluid biomarker that exists. The main challenge arises from the fact that core AD biomarkers are proteins, which can be affected by preanalytical factors such as sample collection conditions, the timing of sample processing, and sample storage conditions [169]. Another limitation is low concentrations of biomarkers in peripheral body fluids. Commonly used detection methods utilize technologies, including immunoblot and electrochemiluminescent immunoassays, and quite often the low concentrations require highly sensitive novel technologies with a lower limit of detection and lower limit of quantification [170]. Besides, the peripheral body fluid biomarker research field has not yet established a consensus on experimental techniques and methods, resulting in low assay standardization [169].

Despite these limitations, the advantages that the novel peripheral body fluid biomarkers possess should be taken into account for easier, faster, and more accessible diagnosis for a wider spectrum of patients. Studies suggested that a combination of multiple biomarkers will improve the diagnostic accuracy when compared with the use of a single biomarker [171,172]. The use of multi-biomarker panels will provide a platform to monitor disease progression longitudinally. Validation of perspective biomarkers in large cohorts of patients would be crucial to be implemented in practical use.

## 5. Conclusions

In combination with clinical examination of cognition and neuropathology, biomarker studies have evolved quickly to understand the pathogenesis and implement early diagnosis for timely therapeutic interventions. Nevertheless, the accessibility to the conventional CSF- and blood-based biomarker tests is hindered due to their invasive and high-cost sampling measures. 

We have also outlined the guidelines for AD diagnosis; however, each study offered its own classification criteria. Therefore, it was difficult to provide an encompassing A/T/N diagnosis status for all the reviewed studies. We hope that the use of A/T/N diagnosis will work in unison with the body fluid biomarkers and provide an overall spectrum of core biomarker modalities in AD. Such elaborated biomarker indices will help better understand the pathology and bolster overall diagnostic accuracy.

This review has explored considerable progress in identifying non-invasive peripheral body fluid biomarkers from nasal discharge, tears, saliva, and urine. A great deal of work on the potential of these non-invasive peripheral body fluid biomarkers has suggested that these biomarkers can be used for early detection and diagnosis of AD and monitoring the disease progression from preclinical to full-blown AD stages. Ideal biomarker characteristics include easy accessibility, high accuracy, minimal invasiveness, cost-efficiency, and reproducibility. A lot of evidence proposed that core AD biomarkers in the non-invasive peripheral body fluid discussed in this review have the possibilities to meet these criteria and be utilized in clinical practice with further research. 

The discovery and application of the non-invasive peripheral body fluid biomarkers may enable early diagnosis, help patient monitoring in clinical trials, or identify disease-relevant molecular pathways to develop novel therapeutic targets. Further refinement in using these biomarkers may lead to the invention of AD screening tests, biosensors, or chip devices with high accuracy and reproducibility. We believe that future studies in this field will undoubtedly have a profound and positive impact on the patients and their families. 

## Figures and Tables

**Figure 1 pharmaceutics-14-01532-f001:**
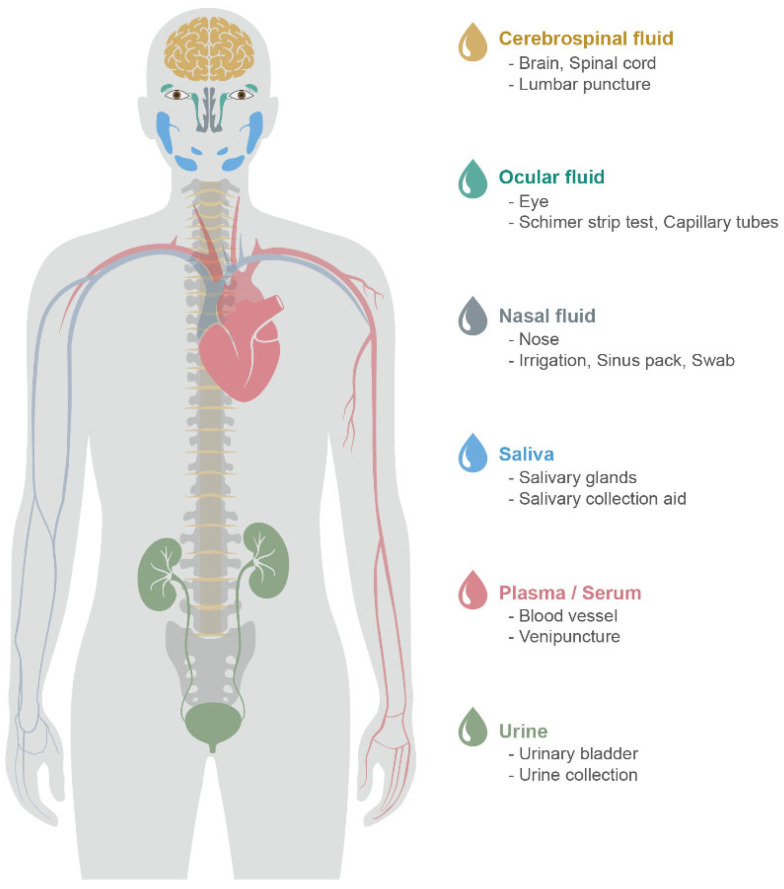
Body fluids for the identification of potential biomarkers for AD. Sample type, organ, sampling method (from above).

**Table 1 pharmaceutics-14-01532-t001:** Clinical Alzheimer’s disease stages-I.

NINDS-ADRDA+DSM-III ^a^ Disease Stage		Dementia				
Diagnostic Subgroups		None	Unlikely	Possible AD ^b^	Probable AD	Definite AD
**Other comments**		−	−	Absence of other diseases capable of producing a dementia syndrome		
**Onset age**		−	Sudden	Atypical	40~90 years	40~90 years
**Neuro-psychological** **test**	MMSE ^c^, Blessed dementia scale, etc.	−	?	+/−	+	+
**Neuroimage**	CT ^d^	−	?	+/−	+	+
**Histology**	Microscopic examination of brain tissue	−	-	−	−	Confirmed by autopsy or biopsy
**Others**	Other signs	−	Focal neurologic signs, seizures, or gait disturbance	−	Cognitive impairments have to be present in two or more areas of cognition	Cognitive impairments have to be present in two or more areas of cognition

^a^ NINDS-ADRDA+DSM: Alzheimer’s Criteria; the National Institute of Neurological Disorders and Stroke and the Alzheimer’s Disease and Related Disorders Association + Diagnostic and Statistical Manual of Mental Disorders; ^b^ AD: Alzheimer’s Disease; ^c^ MMSE: Mini-Mental State Examination; ^d^ CT: Computed Tomography.

**Table 2 pharmaceutics-14-01532-t002:** Clinical Alzheimer’s disease stages-II.

NIA-AA+DSM-V ^a^ Disease Stage			Preclinical	MCI ^b^	AD ^c^ Dementia			Non-AD Dementia
Diagnostic Subgroups		None	Preclinical AD	Possible AD	Mild AD	Moderate AD	Severe AD	OND ^d^
**Neuro-psychological test**	MMSE ^e^ (30~0)	30~25			24~20	19~13	12~	+/−
	CDR ^f^ v1-1993 (0~3)	0	0.5 *questionable*		1	2	3	
	CDR v2-1997 (0~3)	0	0.5 *questionable*	1 *mild CI ^h^*		2 *moderate CI*	3 *severe CI*	
	GDS ^g^ (1~7)	1	2 *very mild**CI*	3 *mild**CI*	4 *moderate**CI*	5, 6 *moderately severe CI*	6, 7 *very severe CI*	
**Neuro-imaging**	amyloid-PET ^i^	−	+	+	+	+	+	−
	tau-PET	−	−	−	+	+	+	+/−
	FDG ^j^-PET	−	−	+/−	+/−	+/-	+/−	+/−
	Structural MRI	−	−	+/−	+/−	+/-	+/−	+/−
**CSF ^k^-biomarker**	CSF Aβ_42_	−	+	+	+	+	+	n/a
	CSF P-tau	−	−	−	+	+	+	n/a
	CSF T-tau	−	−	+/−	+/−	+/−	+/−	n/a

^a^ NIA-AA+DSM: Alzheimer’s Criteria; ^b^ MCI: Alzheimer’s Disease; ^c^ AD: Alzheimer’s Disease; ^d^ OND: Other Neurodegenerative Disease; ^e^ MMSE: Mini-Mental State Examination; ^f^ CDR: Clinical Dementia Rate; ^g^ GDS: Global Deterioration Scale; ^h^ CI: Cognitive Impairment; ^i^ PET: Positron emission tomography; ^j^ FDG: Fluorodeoxyglucose; ^k^ CSF: Cerebrospinal Fluid.

**Table 3 pharmaceutics-14-01532-t003:** Characteristics of ideal body fluid biomarker.

Type	Characteristics	Goals
**Detectability**	Disease specificity	High
	Biomarker sensitivity	High
	Accuracy	High
**Accessibility**	Repeatability	High
	Invasiveness	Low
	Expense	Low
**Stability**	Reproducibility	High
**Reliability**	Early detection	
	Containing pathological correlation	

**Table 4 pharmaceutics-14-01532-t004:** Summary of CSF and blood acquisition procedures.

Body Fluid		Acquisition	Procedure	Reference
**CSF**		Lumbar puncture	1. The subject lies on their side and bends knees toward the chest and chin. 2. An atraumatic spinal needle is injected into the vertebral body L3–L5. 3. CSF is collected in polypropylene tubes about 1–2 mL.	[23]
**Blood**	**Plasma**	Venipuncture	1. Clean the venipuncture site and insert the needle. 2. The blood is collected in blood collection tubes, including anticoagulant (EDTA or heparin). 3. Within 1–2 h, collecting tubes are centrifuged, and then supernatant is transferred into new tubes, including protease inhibitor cocktail.	[33,34]
	**Serum**	Venipuncture	1. Clean the venipuncture site and insert the needle. 2. The blood is collected in blood collection tubes, including a silica clot activator. 3. After clotting for 30 min, samples are centrifuged, and then supernatant is transferred into new tubes, including a protease inhibitor cocktail.	

**Table 5 pharmaceutics-14-01532-t005:** Summary of core AD fluid biomarkers in CSF and blood.

AD Pathology Mechanism	Core Biomarkers	CSF		Blood	
		Change	Reference	Change	Reference
**Aβ pathology**	**Aβ_42_**	Reduced	[29,46,47]	Reduced	[48,49,50]
	**Aβ_42_/Aβ_40_**	Reduced	[51,52,53]	Reduced	[48,50,54]
**Tau pathology**	**P-tau**	Increased	[55,56,57]	Increased	[58,59,60]
	**T-tau**	Increased	[61,62,63]	Increased	[60,64,65]

**Table 6 pharmaceutics-14-01532-t006:** Summary of acquisition procedures for nasal discharge, tear, saliva, and urine.

Body Fluid		Acquisition	Procedure	Reference
Nasal discharge		Nasal irrigation	1. Subjects are comfortably seated, and sterile normal saline (0.9% NaCl) is administered into each nostril. 2. Subjects must close one nostril and then spray or insert sterile normal saline several times into the other nostril. 3. After raising their head slightly back, let the solution stay as if washing the nasal cavities.4. After inserting sterile normal saline, subjects gently blow the nasal discharge fluids into a cup or tube.5. After a few minutes of rest, do the same for the other nostril.6. Samples are stored at −20 °C until use.	[66,67,68]
		Sinus packs	1. Sinus packs or sponges are placed in nasal cavities between the septum and inferior turbinate along the floor. 2. After 1–10 min, the sinus packs or sponges are removed and placed in tubes. In order to retrieve the secretions from sinus packs or sponges, sterile normal saline (0.9% NaCl) solution is added to the tube and stored at 4 °C for about 2 h. 3. The sinus packs or sponges are then placed into a syringe. Mechanical pressure is applied to them by moving the piston action to squeeze the nasal discharge fluid. 4. After the first pressure, the syringe is replaced with a tube and centrifugation is performed to recover all nasal discharge fluids from the sinus packs or sponges. 5. The nasal discharge fluids are then stored at −80 °C for further analysis.	[69]
		Nasal swab	1. Subjects are seated in a comfortable bed, placed in a high fowler’s position in bed, supporting the back of the head. 2. Enter a flexible cotton swab several centimeters with a slow and steady motion along the nose floor. Nasal smears are taken from the inferior concha, middle nasal meatus, olfactory cleft, and common nasal meatus. 3. After resistance is met, rotate the cotton swab several times and withdraw the swab. 4. All cotton swabs are placed in a microtube containing sterile normal saline (0.9% NaCl) for a few minutes, and swabs are removed from the microtube. 5. The solutions are then filtered by centrifugation, and then the filtered solutions are stored at −80 °C until further analysis.	[70,71]
**Tear**		Capillary tube	1. Subjects are seated with their head raised and stimulated by a direct light source or airflow. 2. The reflex tears of the subject are collected with tubes.	[72,73,74]
		Schirmer strip	1. A local anesthetic is needed to collect basal tears, not reflex tears. 2. The bent end of the test strip is placed in the lower eyelid and allowed to absorb the tears for several minutes.	
**Saliva**	**Whole saliva**	Spitting	1. Subjects rinse their mouth and then spit the whole saliva into a sterile tube.	[75,76,77]
	**Submandibular saliva**	Draining	1. To block the opening of parotid ducts and sublingual glands, use cotton gauzes, and to dry up, the floor of the mouse is left. 2. Subjects raise the tongue to open the submandibular gland, and saliva is collected using a disposable pipette.	
	**Sublingual saliva**	Draining	1. To block the opening of parotid ducts and submandibular glands, use cotton gauzes, and to dry up, the floor of the mouse is left. 2. Subjects raise the tongue to open the sublingual gland, and saliva is collected using a disposable pipette.	
	**Parotid saliva**	Draining	1. To collect parotid saliva, parotid cups or collectors are placed, actively stimulating salivary collection.	
**Urine**		Collecting	1. First morning and random collection are not preferred because of increasing variabilities. 2. The mid-stream and second-morning urine is collected in a urine container.	[78,79,80,81]

**Table 7 pharmaceutics-14-01532-t007:** Summary of core AD biomarkers in the olfactory system.

AD Pathology Mechanism	Specimen	Biomarkers	Analytical Method	Results	Reference
**Aβ pathology**	Nasal discharge fluid	Aβ_1–16_	Interdigitated microelectrode biosensor	Increased in AD than in OND and CU ^a^	[105]
	Nasal discharge fluid	Aβ oligomer	Immunoblot	Increased in AD than CU	[67]
	Nasal mucosa by nasal swab	Aβ_42_, Aβ_40_	Immunoassay	No differences in median values between AD and CU	[71]
	Postmortem olfactory epithelium	Aβ	Histopathology	Increased Aβ aggregates in AD patient	[103]
	Postmortem olfactory bulb	Aβ	Histopathology	Increased Aβ load in AD patients	[90,111]
**Tau pathology**	Nasal discharge fluid	T-tau, P-tau	Immunoassay	Positive T- and P-tau in anosmic AD patients	[108]
	Nasal mucosa by nasal swab	T-tau, P-tau	Immunoassay	Positive T- and P-tau in AD patients	[71]
	Postmortem olfactory epithelium	P-tau	Histopathology	Evident PHF-tau ^b^ in AD	[103]
	Postmortem olfactory bulb	P-tau	Histopathology	P-tau deposits in the olfactory bulb of AD patients	[112]

^a^ CU: cognitively Unimpaired; ^b^ PHF-tau: Paired Helical Filament-tau.

**Table 8 pharmaceutics-14-01532-t008:** Summary of core AD biomarkers in tears.

AD Pathology Mechanism	Biomarker	Analytical Method	Results	Reference
**Aβ pathology**	Aβ_42_	Immunoassay	Increased Aβ_42_ levels in AD patients	[118]
**Tau pathology**	T-tau	Immunoassay	Increased T-tau levels in AD patients	[118]

**Table 9 pharmaceutics-14-01532-t009:** Summary of core AD biomarkers in saliva.

AD Pathology Mechanism	Biomarkers	Analytical Method	Results	Reference
**Aβ pathology**	Aβ_42_	Immunoassay	Increased saliva Aβ_42_ levels in mild AD patients	[128,129,130,131]
	Aβ_42_	Magneto-immunoassay	Salivary Aβ_42_ levels increase as the AD severity increases	[132]
	Aβ_42_	Immunoassay	Salivary Aβ_42_ levels were not detectable	[127]
	Aβ_42_	Immunoassay (MILLIPLEX)	Lower salivary Aβ_42_ levels in AD patients	[133]
	Aβ_40_	Magneto-immunoassay	No statistically significant change	[132]
**Tau pathology**	T-tau, P-tau	Immunoassay	Increased P-tau/T-tau ratio in AD patients	[127,134]
	T-tau	Immunoassay	No significant difference in salivary T-tau between AD and healthy control	[135]

**Table 10 pharmaceutics-14-01532-t010:** Summary of core AD biomarkers in urine.

AD Pathology Mechanism	Biomarker	Analytical Method	Results	Reference
**Aβ pathology**	Aβ_42_	Immunoblot	Monomeric Aβ_42_ levels differed according to cognitive impairment	[155]

## Data Availability

Not applicable.

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
