# Peer review of "Non-Invasive Nasal Discharge Fluid and Other Body Fluid Biomarkers in Alzheimer’s Disease"

_pharmaceutics, 2022, doi:10.3390/pharmaceutics14081532_

Round 1

Reviewer 1 Report

The work is very interesting. the chapaters are well build an the information is usefull for the non-invasive diagnosis of Alzheimer disease. please discuss if methods such as miRNA can be used from the samples of fluid you discussed. please chceck: https://pubmed.ncbi.nlm.nih.gov/34357520/. Minor revision.

Reviewer 2 Report

The manuscript (pharmaceutics-1803813) unambiguously describes about peripheral biomarkers, AD diagnosis guidelines, potential of non-invasive bodily fluid biomarkers. Summaries provided in form of the table is another upside of this manuscript. Authors have well described conventional AD body fluid biomarkers and their limitations. Since the main focus of this manuscript is on peripheral body fluid biomarkers which are reviewed only in positive way. While the tables do show the contradiction, for example, table 9 shows Aβ42 inconsistent presence in the saliva of AD patients. As of now there is no reliable peripheral body fluid biomarkers which can be assigned as hallmark for the AD detection. So, there is concern about the limitations to detect the biomarkers in the peripheral body fluids. To improve the manuscript and to balance out between pros and cons, my suggestion is –

 Major Concern-

1.    Authors are suggested to add a section to discuss the limitations with proper references in the detection of the biomarkers from the peripheral body fluids samples.

Reviewer 3 Report

1) In introduction, introduce the problem, motivate the problem and summarize the main contributions

2) No literature survey

3) Show the main limitations in the exsisting works and how the proposed approach overcomes these limitations

4) Include future works

Round 2

Reviewer 2 Report

Thank you for adding the limitation section. No more concerns from my side.